# Effect of the Availability of the Source of Nitrogen and Phosphorus in the Bio-Oxidation of H₂S by *Sulfolobus metallicus*

Javier Silva [1,*], Rodrigo Ortiz-Soto [1], Marjorie Morales [2] and Germán Aroca [3]

[1] Escuela de Ingeniería Química, Pontificia Universidad Católica de Valparaíso, Av. Brasil 2162, Valparaíso 2340025, Chile

[2] Department of Energy and Process Engineering, Faculty of Engineering, Norwegian University of Science and Technology (NTNU), 7491 Trondheim, Norway

[3] Escuela de Ingeniería Bioquímica, Pontificia Universidad Católica de Valparaíso, Av. Brasil 2085, Valparaíso 2340025, Chile

* Correspondence: javier.silva@pucv.cl; Tel.: +56-32-2372618

**Abstract:** The effect of nitrogen and phosphorus availability on the growth of *Sulfolobus metallicus* was analyzed. This archaeon was subjected to a series of nitrogen and phosphorus limitation conditions to determine their effects on growth. The results indicate that *Sulfolobus metallicus* showed a relationship between one of the intermediate oxidation products (tetrathionate) and cell concentration during the exponential growth phase in the absence of nitrogen. Furthermore, significant differences were found in the specific growth rates under different scenarios with ammonia and phosphorus limitation, with values of $0.048\,\mathrm{h}^{-1}$ in the ammonia limitation case. The biomass substrate yield obtained was $0.107\,\mathrm{g_{cel}}\cdot\mathrm{g\,S}^{-1}$. Meanwhile, in the absence of phosphorus, the specific growth rate was $0.017\,\mathrm{h}^{-1}$, and the substrate to biomass yield was $0.072\,\mathrm{g_{cel}}\cdot\mathrm{g\,S}^{-1}$. The results indicate that the ability of *Sulfolobus metallicus* to bio-oxidize H₂S depends on the availability of such nutrients (nitrogen and phosphorus), which affect cellular growth and the types of products generated. This, in turn, influences the oxidation process of various sulfur compounds, resulting in changes in the predominant products formed and the final oxidation of sulfate ions.

**Keywords:** *Sulfolobus metallicus*; nitrogen; phosphorus; nutrients limitation

## 1. Introduction

An extremophile is an organism that lives in extreme conditions, understood as environments very different from those inhabited by most life forms on Earth [1]. These environments include but are not limited to, hot springs, deep-sea hydrothermal vents, salt flats, and even the harsh conditions of outer space [2]. Extremophiles have unique adaptations that enable them to survive and thrive in these environments, such as heat-resistant enzymes that can function at temperatures over 100 °C or survive in highly acidic or alkaline conditions that would be lethal to most organisms [3].

To adapt to extreme environments, extremophiles have developed complex structures and metabolic pathways that allow them to use different energy sources. For example, some extremophiles can use chemosynthesis, which means they use molecules such as sulfur or methane as their energy source instead of sunlight [4]. Additionally, extremophiles have adapted to survive in environments with varying temperatures, pH, redox potential, nutrient availability, and oxygen. The response to these factors differs significantly among organisms. Studying these differences can provide insights into the limits of life on Earth and the potential for life beyond our planet [5].

*Sulfolobus* is a genus of extremophilic microorganisms belonging to the *Sulfolobaceae* family, found to thrive in extreme environments such as thermal acidic waters [6]. Members of this genus are known for their ability to grow aerobically at low pH values and high temperatures, typically in the presence of elemental sulfur [7].

Several studies have been conducted to understand the response of *Sulfolobus* to different conditions and nutrient limitations. For example, Bischof et al. [8] investigated the effect of nitrogen and carbon limitation on the growth of *Sulfolobus acidocaldarius*. They observed transcriptional changes and their subsequent impact on protein production. Similarly, Quehenberger et al. [9] observed changes in lipid composition and the specific growth rate of *Sulfolobus acidocaldarius* when subjected to nutrient limitation. The result suggested that this organism can adapt its metabolism to survive nutrient scarcity.

Osorio and Jerez [10] studied the response of *Sulfolobus acidocaldarius* to a decrease in the source of phosphate, which resulted in a reduction in growth rate due to cellular proteome adaptations. Similarly, the absence of phosphorus can modify the gene expression of *Sulfolobus acidocaldarius*, affecting the regulation of critical metabolic pathways [11]. These studies have provided valuable insights into the unique adaptations of *Sulfolobus* to extreme environments and may have important implications for biotechnology and environmental remediation.

From an industrial standpoint, operating under nutrient limitations due to process restrictions such as environmental conditions or quality control can pose significant challenges. Nutrient limitation can decrease the growth rate of microorganisms, affecting biomass production or metabolites of interest [12]. To optimize bioprocess plants, it is essential to consider the growth conditions that result from nutrient limitation [13].

Kinetic models have been developed to describe the behavior of microorganisms under nutrient-limited conditions. These models consider factors such as the availability of nutrients, the rate of nutrient uptake, and the growth rate of the microorganisms [14]. By incorporating these factors, kinetic models can accurately predict the behavior of microorganisms under different nutrient-limited conditions and can be used to optimize bioprocesses [15]. Growth-limiting conditions should be considered in the development of more accurate kinetic models. In addition, genetic engineering and synthetic biology advances have allowed the design of microorganisms with enhanced nutrient utilization and metabolic capabilities [16]. Optimizing the genetic composition of microorganisms makes it possible to overcome the limitations imposed by nutrient limitation and increase the production of biomass or metabolites of interest [17].

Therefore, considering the effects of nutrient limitation on growth is essential when developing models to describe the behavior of microorganisms in bioprocess plants accurately. By taking a holistic approach to bioprocess optimization, it is possible to overcome the limitations imposed by nutrient limitations and maximize the productivity of bioprocesses [18].

Within the *Sulfolobus* genus, the archaeon *Sulfolobus metallicus* is an irregular coccus with a diameter of 1.5 μm. It is thermoacidophilic, chemolithoautotrophic, and able to grow aerobically in a temperature range of 50 to 75 °C, with an optimum temperature of 70 °C. It thrives in acidic environments with a pH range of 1 to 4.5 and a salt concentration of 0 to 3% [19]. This microorganism can use $Fe^{2+}$ as an energy source, expelling protons into the medium. Its principal use has been bioleaching refractory gold minerals with high-density pulps [20]. Furthermore, *Sulfolobus metallicus* can oxidize total reduced sulfur compounds (TRS) such as hydrogen sulfide ($H_2S$), dimethyl sulfide (DMS), elemental sulfur, and $S_2O_3^{2-}$, as well as other reduced sulfur compounds, as an electron acceptor, with the consequent formation of sulfuric acid as a final product to obtain energy [21,22]. This characteristic has been utilized in biofilters to purify gas streams with high content of odorous substances due to reduced sulfur compounds, resulting in successful removal capacities [23].

The habitats from which *Sulfolobus metallicus* is typically isolated, such as hot springs and acid mine drainage sites, are characterized by low concentrations of compounds that can be used as sources of nitrogen or phosphorus [24,25]. As a result, this microorganism has had to adapt to environments where these nutrients are in low availability [26,27]. In the electron transport chain, electrons produced from the oxidation of sulfur compounds enter at different points depending on their reduction potential, leading to the generation

of ATP, which is used for oxidative phosphorylation coupled to the electron transport chain [28]. Oxygen acts as the final electron acceptor, allowing for $CO_2$ fixation through the 3-hydroxy propionate cycle with high activity of acetyl-CoA and propionyl CoA carboxylases [29]. Since this organism does not have fatty acid chains in its lipid conformation, acetyl-CoA cannot play a role in synthesizing acid chains and must be used in another pathway [30]. For example, in phosphorus-free conditions, the global gene expression of *Sulfolobus acidocaldarius* is altered [31]. In contrast, in this type of microorganism, an ammonium source in the culture conditions positively affects microbial growth and the biological oxidation of reduced sulfur compounds using a sulfur-oxidizing consortium under mesophilic conditions [32]. However, there are no specific reports on the influence of the availability of such essential nutrients as nitrogen and phosphorus on the growth rate of *Sulfolobus metallicus.*

The studies have focused on observing the response of various microorganisms to nutritional constraints, revealing different responses to such conditions. These analyses have examined key metabolites related to the limited nutrient, such as glutamine in nitrogen limitation, ATP in phosphorus limitation, and pyruvate in carbon limitation [33–35].

In addition to observing changes in key metabolites under nutrient limitation, studies have shown that such microorganisms can respond to nutrient constraints by altering their gene expression profiles. For example, nitrogen limitation, nitrogen metabolism, and uptake genes are upregulated, while genes involved in growth and proliferation are downregulated. In phosphorus limitation, phosphate transport and metabolism genes are upregulated, while genes involved in energy-consuming processes are down-regulated [36].

Furthermore, these microorganisms can adapt to nutrient limitations by changing their morphology and cellular composition. Under nitrogen limitation, for example, cells may become smaller and have a higher surface-area-to-volume ratio, which allows for more efficient nutrient uptake. Under phosphorus limitation, cells may increase their membrane phospholipid content and reduce their protein content, which helps to conserve phosphorus [37].

Understanding how these microorganisms respond to different nutrient limitations makes it possible to develop strategies for improving yield and productivity in bioprocesses. For example, adjusting the composition of the growth medium, optimizing feeding strategies, and controlling environmental factors can all be used to improve bioprocess performance. Additionally, understanding the mechanisms underlying nutrient limitations can lead to developing new technologies and bioprocesses that are more efficient and sustainable [38].

The present work aims to study the effect of nitrogen and phosphorus limitation on the growth of *Sulfolobus metallicus* using $H_2S$ as an energy source.

## 2. Materials and Methods

### 2.1. Microorganism and Culture Medium

The purity of the available strain, *Sulfolobus metallicus*, was confirmed by conducting a study on its prokaryotic components using denaturing gradient gel electrophoresis (DGGE), scanning electron microscopy, and X-ray absorption elemental analysis. The study demonstrated that the samples comprised a single cell type with coccoid morphology and an average size of 0.9 μm, which matched the parameters described in the literature for *Sulfolobus* species [39]. Therefore, the analysis confirmed that the strain used in this research was composed of archaea closely related to *Sulfolobus metallicus* and did not contain any contaminating microorganisms.

Additionally, it was found that each sample copy in DGGE showed unique signals (Figure 1) that matched the denaturation point of *Sulfolobus metallicus* DSM 6482, suggesting that the samples used were composed of archaea closely related to *Sulfolobus metallicus*.

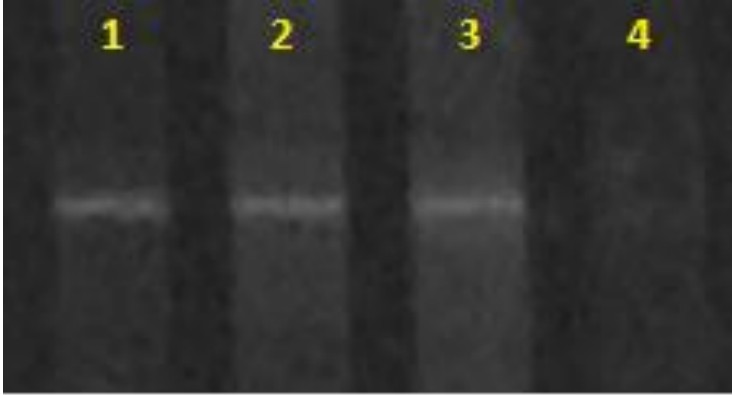

**Figure 1.** DGGE for archaea. The unique signal is observed for each of the samples submitted for analysis. (1) *Sulfolobus metallicus* DSM 6482, (2) and (3) laboratory sample (duplicates), (4) Negative control.

The *Sulfolobus metallicus* strain was cultured in a 250 cm$^3$ Erlenmeyer flask with 5 g·L$^{-1}$ elemental sulfur in Norris [40] medium and placed in a shaker at 70 °C with a rotation speed of 220 rpm. The composition of the Norris medium in g·L$^{-1}$ was as follows: $MgSO_4 \cdot 7H_2O$, 0.5; $(NH_4)_2SO_4$, 0.4; $KH_2PO_4$, 0.2; KCl, 0.1. The initial pH of all media was adjusted to 3.0 using $H_2SO_4$.

### 2.2. Experimental System

Figure 2 shows a diagram of the experimental system. A stirred tank bioreactor with 1 L total volume, 0.48 vvm aeration, and 570 rpm at 70 °C was used, which was continuously fed with a gas stream of 1000 ppm$_v$ of $H_2S$ from a chemical reactor [41]. In addition, a carbon dioxide stream of 5% *v/v* was used to enrich the gas stream.

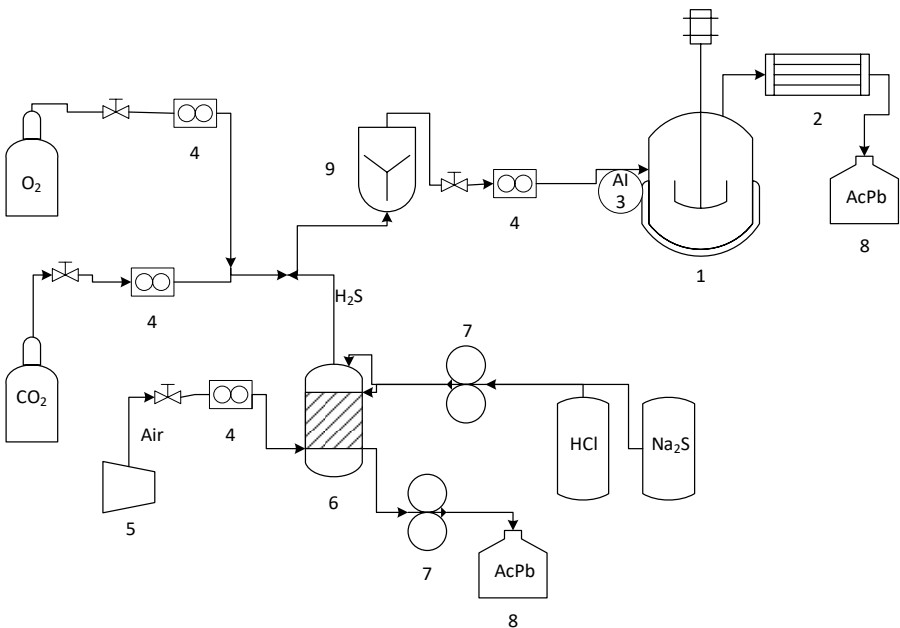

**Figure 2.** Experimental bioreactor (1), condenser (2), $H_2S$ Sensor (3), rotameter (4), compressor (5), chemical reactor (6), a peristaltic pump (7), lead acetate solution (8), mixing column (9).

The gaseous $H_2S$ was generated continuously in the chemical reactor by reacting sodium sulfide solutions ($Na_2S$ 0.13 M) with hydrochloric acid (HCl 0.64 M) using a double-head peristaltic pump. The solutions were stirred using magnetic agitation to promote their contact in the reactor. An airflow regulated by a rotameter with a precision valve was

introduced into the reactor through an inlet on one side. This air stream carried the $H_2S$ generated through the headspace and was removed from the other side of the reactor.

The reactor had two distinct phases: a gaseous phase consisting of the air stream described earlier and a liquid phase comprising the remaining reactor solution. The liquid phase was collected in a plastic drum containing a solution of lead acetate to prevent the release of toxic gas into the environment. The lead acetate solution reacts with hydrogen sulfide to form galena and acetic acid. Next, the outlet stream from the chemical reactor was mixed with a 5% $v/v$ carbon dioxide stream and fed through a gas cylinder. Finally, the gas stream was divided into two feed streams: one for the bioreactor and one for the hydrogen sulfide sensor, which was manually activated using a needle valve.

A series of experiments was conducted to investigate the impact of ammonium and phosphate availability on the growth of *Sulfolobus metallicus*. First, to examine the effect of ammonia, cultures were prepared using a modified Norris medium with a pH of 3.0 and containing 0.4 g·L$^{-1}$, 0.2 g·L$^{-1}$, and 0 g·L$^{-1}$ of $(NH_4)_2SO_4$. Similarly, to study the effect of phosphate, cultures were prepared using a modified Norris medium with a pH of 3.0 and containing 0.2 g·L$^{-1}$, 0.1 g·L$^{-1}$, and 0 g·L$^{-1}$ of $KH_2PO_4$. In each case, triplicate measurements of cell concentration, tetrathionate, elemental sulfur (as intermediate byproducts), and sulfate, ammonium, and phosphate levels were taken.

### 2.3. Analytic Techniques

The ammonium concentration was determined using the phenol-hypochlorite method [42]. Next, the cell concentration was quantified in triplicate using a Petroff Hauser chamber and a fluorescence microscope with phase contrast (Nikon model DS-Fi1, Tokyo, Japan). Finally, the samples were observed and counted using a Carl Zeiss microscope, model Axiostar Plus (Jena, Germany).

The sulfate concentration was determined using the turbidimetric suspension method [43]. This method added barium chloride to the sample, which reacted with sulfate ions to form insoluble barium sulfate. Finally, the sulfate concentration was determined by measuring the turbidity of the solution using a spectrophotometer (Jenway brand, model 6405UV/Vis, Vernon Hills, IL, USA).

The tetrathionate concentration was determined using the cyanolysis technique [44]. This involves reacting tetrathionate with potassium cyanide to form a colored product upon adding ferric nitrate solution. First, the concentration was measured by considering an absorption reading of 460 nm. Next, the elemental sulfur was determined using a colorimetric method [45].

Linear regression analysis with a 95% confidence level was used to analyze the results. An F-test was conducted to assess the homogeneity of variances in the cell counts among the groups. Paired two-sample *t*-tests with the same confidence level were used to determine significant differences in cell growth behaviors between each group.

### 3. Results

### 3.1. Study of the Nitrogen Limitation

Figure 3 displays the variations in the concentrations of cells, tetrathionate, sulfur, sulfate, and ammonia over time for cultures with ammonia feed concentrations of 0.4, 0.2, and 0 g·L$^{-1}$. These sulfur compounds are considered target analysis compounds because they are produced during the oxidation of hydrogen sulfide in the metabolic pathway of *Sulfolobus*, which utilizes hydrogen sulfide as an energy source. The oxidation process involves a series of reactions that transfer electrons from the hydrogen sulfide molecule to oxygen or other electron acceptors [46].

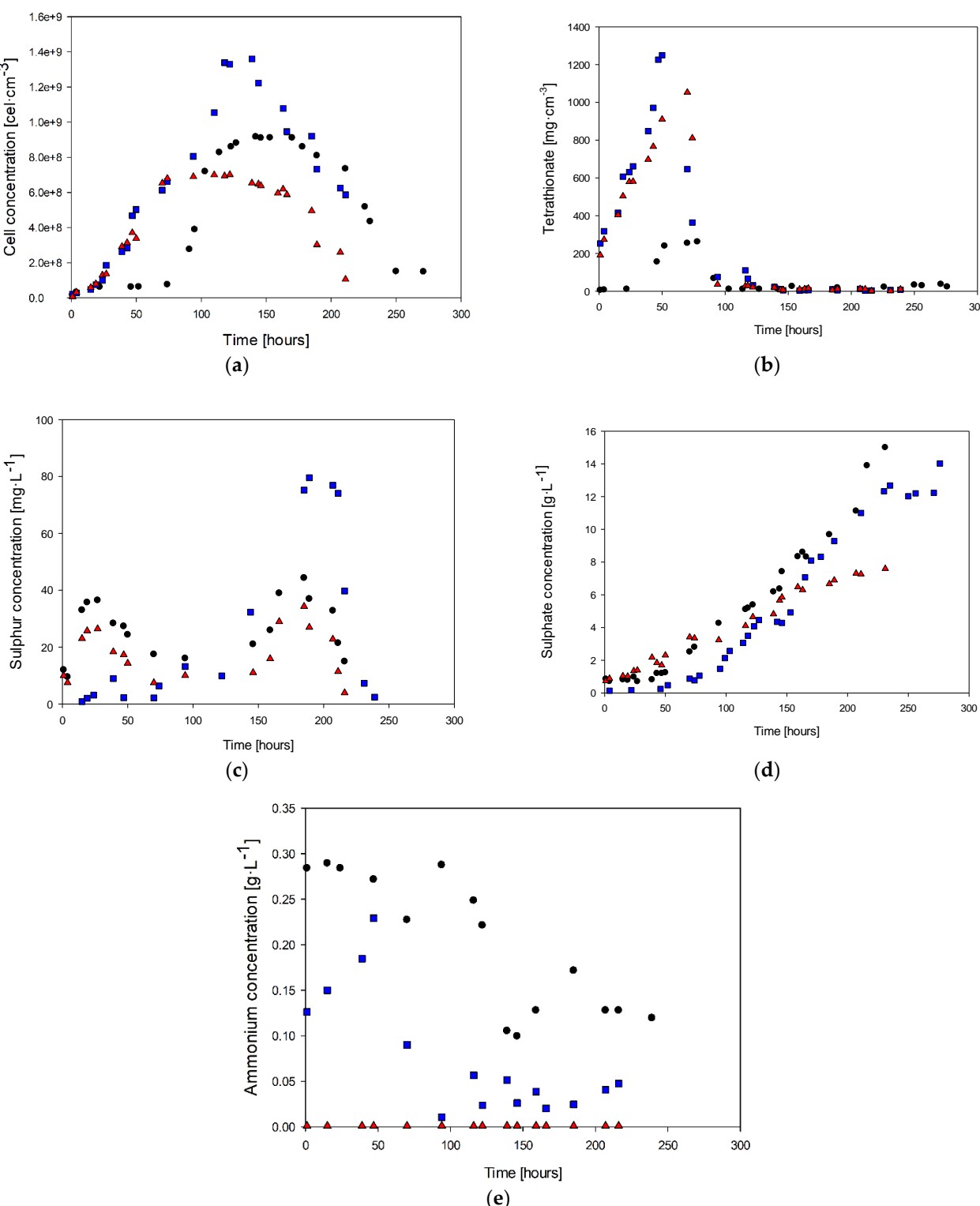

**Figure 3.** Cell concentration (**a**), tetrathionate (**b**), sulfur (**c**), sulfate (**d**), and ammonia (**e**): (●) Culture with 0.4 g·L$^{-1}$ of ammonia, (■) culture with 0.2 g·L$^{-1}$ of ammonia, (▲) Culture without ammonia.

Sulfate is the end product of the complete oxidation of hydrogen sulfide, which is produced from the oxidation of intermediate compounds such as sulfite and thiosulfate. Tetrathionate is produced from thiosulfate and elemental sulfur oxidation and can oxidize to sulfate. Elemental sulfur is also made from the oxidation of hydrogen sulfide and can be further oxidized to tetrathionate or sulfate, depending on the environmental conditions [29].

The generation of sulfate, tetrathionate, and elemental sulfur reflects the diversity of sulfur compounds that *Sulfolobus* can use as energy sources and the complexity of the metabolic pathways involved in the oxidation of hydrogen sulfide [47]. This highlights the remarkable adaptability and versatility of *Sulfolobus* in thriving in extreme environments by efficiently utilizing the available sulfur compounds.

Figure 3a shows the biomass behavior for the three study cases. It is possible to observe that in the culture with $0.4$ $g \cdot L^{-1}$ of ammonia in the feed, at 70 h, the cell concentration was $7.6 \times 10^7$ cells$\cdot$mL$^{-1}$, whereas, in the cultures with 0.2 and 0 $g \cdot L^{-1}$ ammonium, the biomass concentrations were $6.1 \times 10^8$ $g \cdot L^{-1}$ and $6.5 \times 10^8$ $g \cdot L^{-1}$, respectively. The maximum cell concentrations were $9.17 \times 10^8$, $1.36 \times 10^9$, and $7.02 \times 10^8$ $g \cdot L^{-1}$ for each study case at 142, 139, and 122 h, respectively. On the other hand, the beginning of the exponential phases were 52, 19, and 19 h, respectively.

Table 1 presents the results of the statistical tests conducted on the cell concentration data at different time points for each culture condition. The results show that the pairs $0.4–0.2$ $g \cdot L^{-1}$ and $0.4–0$ $g \cdot L^{-1}$ have equal variances, whereas the pair $0.2–0$ $g \cdot L^{-1}$ has different variances.

**Table 1.** Statistical parameters for analysis of the effect of the ammonia in the Culture of *Sulfolobus metallicus*.

| | Ammonia in the Culture Medium | | |
|---|---|---|---|
| **Couple** | **0.4–0.2 $g \cdot L^{-1}$** | **0.2–0 $g \cdot L^{-1}$** | **0.4–0 $g \cdot L^{-1}$** |
| F | 1.53 | 3.16 | 2.08 |
| F-critic | 2.04 | 1.97 | 1.98 |
| *p*-value (F) | 0.16 | 0.01 | 0.04 |
| Statistic t | −1.10 | 2.11 | 0.98 |
| t-critic | 2.01 | 2.03 | 2.02 |
| *p*-value (*t*-test) | 0.28 | 0.04 | 0.33 |

The results suggest no significant difference in the cell growth behavior between the $0.4–0.2$ $g \cdot L^{-1}$ and $0.4–0$ $g \cdot L^{-1}$ ammonium groups. However, for the $0.2–0$ $g \cdot L^{-1}$ ammonium couple, the *p*-values were less than 5%, indicating a significant difference in cell growth behavior. In addition, the exponential phase onsets were significantly different in the 0–0.4 case. The lack of significance in the complete culture for this pair may be due to the scattering of data in counting techniques. In summary, the results suggest that the absence of ammonium significantly affects cell growth behavior.

Table 2 displays the calculated parameters during the exponential growth phase under various nitrogen source availabilities and their corresponding confidence intervals. The results reveal that a higher specific growth rate was attained at higher ammonia concentrations, highlighting the nutritional importance of ammonia for the microorganism. Nevertheless, the findings also suggest that although ammonium impacts the growth of the microorganism, it does not produce a significant effect, as indicated by the slight variance observed from 0.4 to 0.

**Table 2.** Growth parameters for each reactor in the study of the effect of the nitrogen limitation.

| | Ammonia in the Culture Medium | | |
|---|---|---|---|
| **Growth Parameter** | **0.4 $g \cdot L^{-1}$** | **0.2 $g \cdot L^{-1}$** | **0 $g \cdot L^{-1}$** |
| $\mu$ [h$^{-1}$] | $0.068 \pm 0.012$ | $0.052 \pm 0.011$ | $0.048 \pm 0.007$ |
| $Y_{X/S}$ [gcel$\cdot$gS$^{-1}$] * | $0.221 \pm 0.005$ | $0.161 \pm 0.006$ | $0.102 \pm 0.009$ |
| $Y_{X/S}$ 1 [gcel$\cdot$gS$^{-1}$] ** | $0.191 \pm 0.009$ | $0.092 \pm 0.004$ | $0.070 \pm 0.006$ |

* Determined by sulfate concentration. ** Determined by substrate balance.

Furthermore, the ammonia concentration in the culture medium affects the ability of the microorganism to oxidize hydrogen sulfide, which in turn impacts the biomass

substrate yield ($Y_{X/S}$). The confidence intervals indicate a significant improvement in the yield at higher ammonia concentrations, implying better utilization of the energy source due to the increased nutrient levels of ammonia.

The tetrathionate (Figure 3b) exhibited a significant increase in concentration before the beginning of the next biomass production stage, as it is an intermediate product. After the first 50 h of culture, the concentration of tetrathionate was 240, 580, and 630 mg·$L^{-1}$ for each study case. Subsequently, the highest concentration of tetrathionate was observed during the exponential growth phases, where this first intermediate oxidation product was formed, reaching values of 262, 1052, and 1250 mg·$L^{-1}$, respectively. Finally, an abrupt decrease in the tetrathionate concentration was observed due to the cessation of cell growth as the culture reached a steady state. Conversely, the total consumption of tetrathionate indicates the onset of the death cell phase.

The other quantified oxidation intermediate product, elemental sulfur, showed a similar trend to tetrathionate (Figure 3c), with an initial concentration followed by consumption once the stationary growth phase was reached. However, the sulfur concentrations were 95%, 77%, and 70% lower than the tetrathionate values achieved at the final production before reaching the steady state. When comparing the tetrathionate case with the elemental sulfur behavior, the highest concentration (80 mg·$L^{-1}$) was achieved around 200 h for the case of 0.2 g·$L^{-1}$ of ammonia in the feed. This concentration was reached during the decay phase when all the tetrathionate was oxidized to elemental sulfur. Elemental sulfur production was observed in the first 27 h and consumed during the next 70 h. The deceleration phase coincides with the consumption of elemental sulfur, and the sulfur concentration remained constant during the steady stage (between 70 and 150 h). Finally, sulfur presented the highest concentration in the second curve of production and consumption with the cell decay stage (after 150 h), which was observed due to sulfur oxidation to the final product (sulfate ion).

Figure 3d shows the sulfate concentration along the different cases of cultures. In the first 50 h of culture, when the highest concentration of tetrathionate was reached, there was no significant increase in the sulfate ion concentration, which was the time when sustained production of this compound began. After that, the sulfate concentration showed a cumulative behavior related to biomass production during the entire culture time. Considering the $H_2S$ oxidation to sulfate ion as the final product, it is possible to determine a production rate of 0.06 g·$L^{-1}$·$h^{-1}$, which is 50% higher than the culture medium without ammonia.

Similarly, ammonium concentration only decreased at the end of the culture when biomass production slowed down (Figure 3e). In the case with the highest ammonia feed, the ammonium concentration was 0.28 g·$L^{-1}$ during the cell growth and stationary phases. After this period, there was no further consumption of this nitrogen source. During the exponential growth stage, there was an increase in ammonia concentration. When the steady state was reached (at 70 h), the ammonia concentration decreased to values of 0.04 g·$L^{-1}$ for the case with 0.2 g·$L^{-1}$ of ammonia in the feed.

### 3.2. Study of the Phosphorous Limitation

Figure 3 summarizes the results of the reactors fed with 0.2, 0.1, and 0 g·$L^{-1}$ $KH_2PO_4$ showing the concentrations of cells, tetrathionate, elemental sulfur, sulfate, and phosphate.

Figure 4a shows the cell concentration over time in the study of the absence of phosphate. It is possible to observe that at 94 h of culture, the cell concentrations were $3.89 \times 10^8$, $5.14 \times 10^8$, and $6.09 \times 10^8$ with 0.2, 0.1, and 0 mg·$L^{-1}$ of phosphate in the feed, respectively. The maximum cell concentrations were $9.17 \times 10^8$, $9.11 \times 10^8$, and $6.75 \times 10^8$ at 142, 170, and 153 h for each case study. The lag phase duration reduced over time, presenting values of 60, 5, and 5 h in each case, respectively. Table 3 shows the results of the statistical test for each analyzed pair. The F-test indicates that only the couple 0.2–0 g·$L^{-1}$ has significant differences in the variance. However, every *t*-test suggests no significant difference in the cell count curve considering all culture periods due to the absence of the phosphorus source.

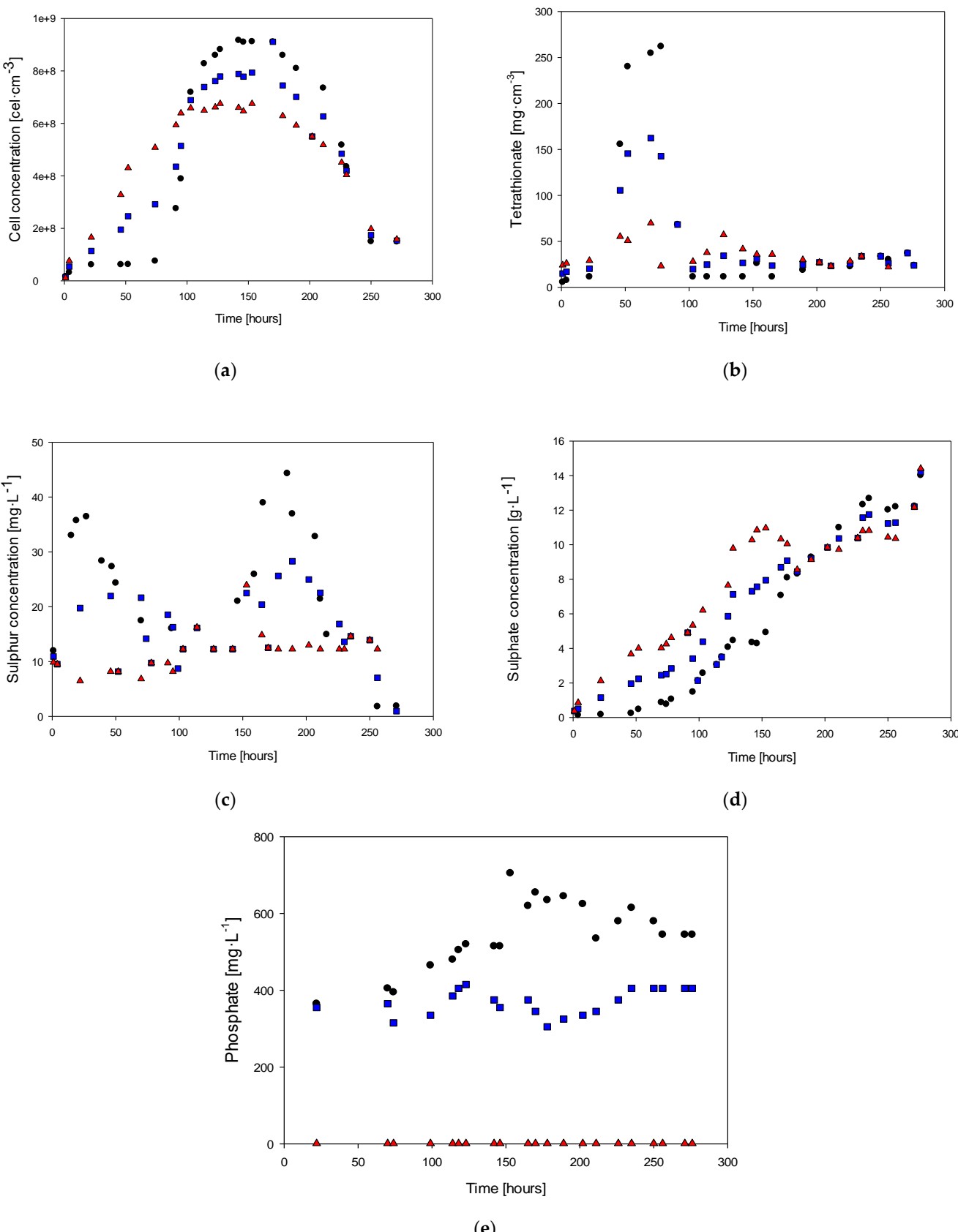

**Figure 4.** Cell concentration (**a**), tetrathionate (**b**), elemental sulfur (**c**), sulfate (**d**), and phosphate (**e**): (●) culture with 0.2 g · L$^{-1}$ of phosphate, (■) culture with 0.2 g · L$^{-1}$ of phosphate, (▲) culture without phosphate.

**Table 3.** Statistical parameters for analysis of the effect of the phosphorus in the Culture of *Sulfolobus metallicus*.

| Couple | Phosphorus in the Culture Medium | | |
| --- | --- | --- | --- |
| | 0.2–0.1 g·L$^{-1}$ | 0.1–0 g·L$^{-1}$ | 0.2–0 g·L$^{-1}$ |
| F | 1.80 | 1.61 | 2.89 |
| F-critic | 2.04 | 2.05 | 2.05 |
| *p*-value (F-test) | 0.09 | 0.14 | 0.01 |
| Statistic t | 0.24 | 0.10 | 0.34 |
| t critic | 2.01 | 2.02 | 2.03 |
| *p*-value (*t*-test) | 0.81 | 0.92 | 0.73 |

Table 4 presents the calculated parameters and their confidence intervals for different levels of phosphorus availability during the exponential growth phase. The higher specific growth rate was observed at null phosphorus concentrations, suggesting inhibitory effects at the concentrations used. The confidence levels indicate no significant differences in the specific growth rates between the 0.2 g·L$^{-1}$ and 0.1 g·L$^{-1}$ phosphorus concentration cases due to overlapping ranges. Although the *t*-test results do not show significant differences between the cultures, these differences may be attributed to changes in cultural behavior, specifically in the exponential growth phase. Conducting a *t*-test solely on the exponential growth phase for the 0.2 g·L$^{-1}$–0.1 g·L$^{-1}$ couple yields a *p*-value of 3.3%, confirming that a difference exists due to the absence of phosphate during this phase.

**Table 4.** Growth parameters for each reactor in the study of the effect of the phosphorus limitation.

| Growth Parameter | Phosphate in the Culture Medium | | |
| --- | --- | --- | --- |
| | 0.2 g·L$^{-1}$ | 0.1 g·L$^{-1}$ | 0 g·L$^{-1}$ |
| $\mu_1$ [h$^{-1}$] | $0.052 \pm 0.016$ | $0.035 \pm 0.002$ | $0.017 \pm 0.004$ |
| $Y_{X/S}$ [gcel·gS$^{-1}$] * | $0.221 \pm 0.013$ | $0.38 \pm 0.005$ | $0.072 \pm 0.012$ |
| $Y_{X/S}$ 1 [gcel·gS$^{-1}$] ** | $0.196 \pm 0.012$ | $0.30 \pm 0.012$ | $0.059 \pm 0.009$ |

* Determined by sulfate concentration. ** Determined by substrate balance.

The results indicate that in the absence of phosphorus, the microorganism is expected to increase its metabolic capacity in terms of growth rate. However, the yield decreases, suggesting that the oxidative capacity of the microorganism decreases in the absence of phosphorus in the culture medium.

Figure 4b shows the tetrathionate concentration over time in studying the effect of the phosphate absence. It indicates that after the lag phase, tetrathionate production began and reached a maximum concentration of 160 mg·L$^{-1}$, for the case of 0.1 mg·L$^{-1}$ of phosphate in the fed, to be consumed during the stationary phase.

Figure 4c shows that the concentration of elemental sulfur ranged from 5 to 45 mg·L$^{-1}$. During the lag phase, there was a decrease in the sulfur concentration, while there was no production of tetrathionate or sulfate ions.

The sulfate concentration (Figure 4d) showed a similar trend to the cell growth plot due to the lag phase, with sulfate production being 0.7 g·L$^{-1}$ in the lag phase and increasing continuously during the exponential growth phase.

Figure 4e shows that the phosphate concentration increased during the exponential growth phase, which suggests that the microorganism was producing phosphate as a byproduct of its metabolic functions. The rate of phosphate production was lower when less phosphate was fed to the culture, which suggests that the microorganism could regulate its phosphate production based on the availability of phosphate in the culture medium. When phosphate was not fed, there were no significant changes in the phosphate concentration, suggesting that the microorganism could not produce phosphate without an external source of this nutrient.

The sulfate ion concentration peaked at 14 g·L$^{-1}$ and followed the same trend as in previous experiments, attributed to the continuous oxidation of H$_2$S. However, under the new conditions with different levels of phosphorus availability (0.1 and 0 mg·L$^{-1}$), the cell concentration was 15% lower than in the absence of phosphorus, but the dynamic curves were distinct. This suggests that the presence of phosphorus in the culture medium positively affects the growth of the microorganism, while its absence leads to a decrease in cell concentration. On the other hand, the absence of phosphorus did not significantly affect the sulfate ion concentration, as it continued to increase due to the continuous oxidation of H$_2$S.

The concentrations of tetrathionate and elemental sulfur had reductions of 57% and 85%, respectively, when the reactor was fed with 0.1 g·L$^{-1}$ KH$_2$PO$_4$ compared to when it had an absence of phosphate. Significant production and consumption of tetrathionate were not observed, and the concentration of elemental sulfur ranged from 0.7 to 2.5 mg·L$^{-1}$. This implies that in the absence of phosphorus in the culture medium, the elemental sulfur is oxidized as soon as it is produced. The results suggest that the presence of phosphorus in the culture medium plays an important role in the metabolism of elemental sulfur by the microorganism. In the absence of phosphorus, the elemental sulfur produced was rapidly oxidized, leading to lower concentrations of tetrathionate and elemental sulfur than when the reactor was fed with 0.1 g·L$^{-1}$ KH$_2$PO$_4$. This may be because phosphorus is a key nutrient in many metabolic processes, including sulfur metabolism. The results also suggest that the microorganism may have a lower oxidative capacity in the absence of phosphorus, as evidenced by the decrease in yield.

## 4. Discussion

The initial presence of tetrathionate during the early stages of the culture indicates that the microorganism is using it as an energy source for growth and metabolism, as it is an intermediate compound produced from the oxidation of the sulfide ion. As the concentration of tetrathionate decreases, the microorganism shifts to oxidizing other sulfur compounds, including elemental sulfur, until ultimately oxidizing sulfate as the final product [48].

In *Sulfolobus metallicus*, the oxidation of H$_2$S usually results in the formation of elemental sulfur as an intermediate compound. However, when ammonia is present, it can affect the pathway of sulfur compound oxidation. Notably, the production-consumption curve of elemental sulfur was only observed during the oxidation of tetrathionate. This implies that ammonia may form different intermediates or products, leading to an altered pathway to oxidize sulfur compounds [49].

The presence of ammonia in the culture caused a different behavior pattern compared to the culture without ammonia. The results showed that ammonia was consumed during the cell growth and stationary phases, indicating that cell death occurred due to the accumulation of an inhibitory product or changes in the physicochemical environment. It is also possible that there was a lack of another nutrient besides the continuously fed energy source, as the nitrogen source was not fully consumed. Further investigation is necessary to identify the specific cause of cell death in the presence of ammonia.

The study conducted by Osorio and Jerez [10] investigated the adaptive response of the archaeon *Sulfolobus acidocaldarius* to phosphorus-limiting conditions. The results showed that growth was affected when cells were subjected to phosphorus limitation in batch culture. Similarly, in the present study, the absence of a phosphorus source decreased the cell growth rate compared to when phosphate was present in the medium, also observed in *Sulfolobus caldarius*. Furthermore, in the absence of nitrogen, genes involved in amino acid transport and metabolism are upregulated [50], whereas in the absence of phosphorus, genes involved in phosphate transport and metabolism are upregulated. Furthermore, the absence of nitrogen or phosphorus also increased stress response-related genes, indicating that the cells were experiencing stress due to nutrient limitation [51].

Without oxygen, *Sulfolobus metallicus* can switch from oxidative phosphorylation to anaerobic respiration using electron acceptors such as elemental sulfur, sulfate, and metals. The metabolic response to metal-reducing conditions involves a shift in carbon metabolism from the tricarboxylic acid (TCA) cycle to the 3-hydroxy propionate cycle for carbon fixation. This adaptation allows *Sulfolobus metallicus* to produce energy and reduce metals simultaneously, highlighting the metabolic versatility of this organism [27].

Nemati et al. [52] reported a significant decrease in the cell size of *Sulfolobus metallicus* under stress conditions, where the cells exhibited a reduction in diameter by 25% and a corresponding reduction in volume by 50%. This response was attributed to changes in the cell wall structure and composition, which may represent a mechanism for the cells to adapt to stressful conditions. Figure 5 shows an electron micrograph obtained from *Sulfolobus metallicus* (green arrows) where its coccoid shape can be observed; they are gram-negative irregular, about 1.5 µm wide in their regular shape while at the same time showing coccoids with certain irregularities with an average size of 0.9 µm.

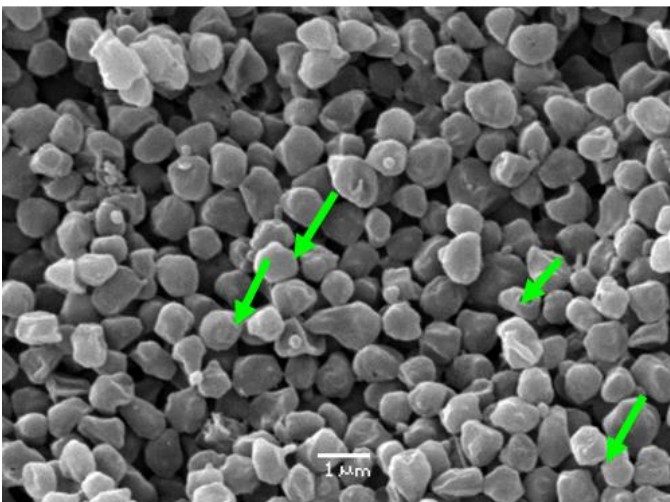

**Figure 5.** Electron microscopy of suspension of *S. metallicus* (green arrows). Magnification: 10,000×.

The absence of a phosphorus source in experiments could affect the viability of the microorganism during the growth phase by reducing the efficiency of energy utilization from the oxidation of sulfur compounds or by causing the accumulation of toxic metabolites [53]. In the absence of phosphorus, the microorganism may not be able to efficiently utilize the energy available from the oxidation of sulfur compounds, which can lead to reduced growth rates and even cell death [54]. Furthermore, the accumulation of toxic metabolites can also negatively impact the viability of the microorganism [55]. Thus, the absence of intermediate oxidation products suggests that the microorganism efficiently utilizes energy and prevents the accumulation of toxic metabolites, which could otherwise negatively impact its viability [56].

The biomass substrate yield, under phosphorus-free conditions, was lower than when phosphate was added or when Norris complete medium was used; however, the results could indicate the presence of an optimal value for phosphorus concentration from a yield perspective. The results suggest that phosphorus is an essential nutrient for the substrate yield and specific growth rates of *Sulfolobus metallicus*. The biomass substrate yield was lower when phosphate was not added, indicating the importance of phosphorus as a nutritional requirement. However, there may be an optimal phosphorus concentration for the substrate yield, as noted in the results in Table 4. Overall, the results suggest that maintaining an optimal phosphorus concentration is essential for optimizing the substrate yield and specific growth rates of *Sulfolobus metallicus*.

## 5. Conclusions

The bio-oxidation capacity of hydrogen sulfide by *Sulfolobus metallicus* is highly dependent on the availability of nutrients such as nitrogen and phosphorus in the culture medium. Researchers have observed that the concentration of phosphorus and nitrogen in the culture medium positively affects the growth and oxidation of $H_2S$ by *Sulfolobus metallicus*. Specifically, increasing the concentration of phosphorus and nitrogen in the culture medium has resulted in significant improvements in specific growth rate and yield, indicating the importance of nutrient availability in bioprocesses involving *Sulfolobus metallicus*.

From an energy source standpoint, researchers have observed the sequential oxidation of several sulfur compounds from $H_2S$ according to their oxidation state. Furthermore, these oxidations have been found to anticipate the different cell growth phases, highlighting the complexity of the metabolic pathways involved in the oxidation of $H_2S$ by *Sulfolobus metallicus*.

Overall, these findings have important implications for optimizing bioprocesses involving *Sulfolobus metallicus*. By ensuring the availability of key nutrients and understanding the complex metabolic pathways involved in $H_2S$ oxidation, it is possible to maximize the bio-oxidation capacity of *Sulfolobus metallicus* and increase the productivity of bioprocesses.

**Author Contributions:** Conceptualization, J.S.; methodology, G.A.; software, J.S.; validation, J.S., M.M.; formal analysis, J.S.; investigation, M.M.; resources, G.A.; data curation, J.S.; writing—review and editing, R.O.-S.; visualization, R.O.-S.; supervision, J.S.; project administration, G.A.; funding acquisition, G.A. All authors have read and agreed to the published version of the manuscript.

**Funding:** National Agency of Research and Development (ANID) Ministry of Science Knowledge and Innovation of Chile, Project FONDECYT 1211569.

**Institutional Review Board Statement:** Not applicable.

**Informed Consent Statement:** Not applicable.

**Data Availability Statement:** The data that support the findings of this study are available on request from the corresponding author.

**Conflicts of Interest:** The authors declare no conflict of interest. The funders had no role in the study's design, in the collection, analyses, or interpretation of data, in the writing of the manuscript, or in the decision to publish the results.

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
