# Peer review of "Effect of the Availability of the Source of Nitrogen and Phosphorus in the Bio-Oxidation of H2S by Sulfolobus metallicus"

_fermentation, doi:10.3390/fermentation9050406_

Round 1
Reviewer 1 Report
Title: Effect of the availability of the source of nitrogen and phosphorus in the bio-oxidation of H2S by Sulfolobus metallicus
The authors present in this work the effect of nitrogen and phosphorus on the growth of Sulfolobus metallicus of using H2S as energy source. This work could be of interest to people in the field and deserves being published. In general, English style must be revised in means of being clear. Some of the aspects that must be revised are the following.
Line 14-16 : “The results indicate that Sulfolobus metallicus, in the absence of nitrogen, showed a relationship between one of the intermediate oxidations (tetrathionate) and cell concentration during the exponential growth phase.”
Please, reformulate this sentence: do you mean an intermediate chemical obtained by oxidation(tetrathionate) or a chemical intermediate oxidation reaction (in which case you should indicate which one are you referring to)?
Line 21: “the biooxidation ability of Sulfolobus metallicus for bio-oxidizing H2S”, the second term is redundant
Line 32: but are not limited to, The comma is not needed here.
Line 68
“In order to optimize bioprocess plants, it is essential to consider the kinetic limitation conditions that result from nutrient limitation”
Please, avoid repeting words.
Line 100. Please, mention some of these environments.
Line135-6 “as it can help inform strategies for improving yield and productivity”, please reformulate this sentence.
MM section
Please, include growth conditions (e.g., temperature, stirring) in 2.1 section when strain is growing in flasks. Or is this strain always grown in the bioreactor? If so, please, specify it and avoid using the word “flasks”.
Figure 1 is not mentioned anywhere in the text, please, introduce a mention in 2.2 section. When describing the bioreactor. I would suggest to briefly explain some of the components in Figure 1, eg. “lead acetate is used for detection of hydrogen sulfide as previously described (a proper ref as the next one or whatever you use).”
HUNTER, C.A. and CRECELIUS, H.G. Hydrogen sulfide studies: detection of hydrogen sulfide in cultures. Journal of Bacteriology, 1938, vol. 35, p. 185.
It would be nice also to include as supplementary, pictures of the bioreactor built.
Line 155: “cultures were carried out with 0.4, 0.2, and 0 g·L-1 of (NH4)2SO4…”
As Norris medium initially contains 0.4 of this compound, are you using a modified Norris minimal medium? The same happens with KH2PO4.
Section 2.3. Although the techniques used are already published, it would be convenient to briefly described in which consists of as it is annoying for the reader searching for each method in different papers.
Results
Figure 2. It is not clear at which time after the inoculum of the bioreactor the experiments are done. What is consider time 0? On the other hand, in my pdf, numbers and letters of figure 2a are blurred, more than the rest.
Line 173, A brief explanation must be given about why authors have considered to evaluate the evolution of these compounds (tetrathionate, sulfur, sulfate) and no other intermediate byproducts. It would be of interest to mention in the introduction the main mechanisms or reactions that form these intermediates (maybe with a supp. Figure) so it can be clear the relationships among them and the object of study.
Line 186-190. I consider that the explanation of the statistical methods should be moved to MM section.
Line 258 “Figure 3a shows the behavior of the cell concentration”
I found the word “behavior” in this sentence a bit strange. In fact, Figure 3a displays concentration of cells vs time and phosphate presence in the medium. It is the same in line 286, I would avoid this term if possible in the context it has been used.
Line 261 “case study”, please correct.
“The lag phase was reduced at lower phosphate fed at 60, 5, and 5 h, respectively.”
I am sorry I do not understand well this sentence. was it reduced from 60 to 5 h or what do you really want to mean?
Line 264 “…has significant variance differences in the variance.”
Line 276-278 “A t-test developed considering only the exponential growth phase gives a p-value of 3.3%, confirming that a difference exists because of the absence of phosphate in such a phase”
I would add this analysis made with data of exp. phase as supplementary as well as indicate in table 3 and 1 which data have you considered for the analysis (for instance, which period of times).
Line 326. “Overall, these findings provide insights into the metabolic mechanisms of elemental sulfur oxidation by microorganisms and the role of phosphorus in this process”.
This sentence must be removed from results to discussion.
Line 332. “Tetrathionate is an intermediate compound that forms during the oxidation of sulfide ions by Sulfolobus metallicus.”
Do you mean “that is formed”?? As I have suggested before, it would be useful to count with a figure that displays the mentioned compounds reactions.
Line 346. “suggesting that cell death occurred due to the accumulation of an inhibitory product or changes in the physicochemical environment.”
Have you analyzed the cell medium to detect accumulation of any toxic compounds? What kind of inhibitory compounds could be responsible for the cell death? The physicochemical parameters are controlled in the reactor, have you detected any imbalance? Could be possible that a cofactor necessary for the metabolism could be missing or reduced?
Line 356-360. This paragraph mentions what it has been reported before about size. Have you observed under microscope the size or form of the cells after the treatments? I can not see otherwise why you mention this fact in the discussion section. It would be interesting to contrast this data with your experiments.
Line 379- “The oxidation kinetics of Sulfolobus metallicus presented sequential oxidation of various sulfur compounds according to their oxidation state…”
I cannot find data about the oxidation kinetics in the article, just growth data, but not kinetic parameters (proper from chemical reactions, Vmax, Km and so on) are provided.
On the other hand, in the introduction, authors mention that the absence of phosphate or nitrogen have been correlated with genetic expression modifications in this strain, are the genes involved in the oxidation processes mentioned in this work among them? Maybe this can be discussed.
Discussion
On the light of data provided, I would include the best conditions of Sulfolobus metallicus growth for this kind of bioreactor.
References
There are some format mistakes, please revise.
As a few examples: line 430 Environ. Microbiol., 2020, 22 (the first “,” is redundant)
Line 447 Watanabe, Y.-i.” (-i¿?)
Line 498 Escherichia coli (italics missing) and so on.
Author Response
In the attached document you will find the answers to the observations made by the reviewer.

Reviewer 2 Report
The manuscript “Effect of the availability of the source of nitrogen and phosphorus in the bio-oxidation of H2S by Sulfolobus metallicus” by Javier Silva, Rodrigo Ortiz-Soto, Marjorie Morales and Germán Aroca, submitted to Fermentation journal to section "Microbial Metabolism, Physiology & Genetics", within the Special Issue “Application of Extremophiles in Biological Degradation and Conversion” is very interesting and fits well in this SI.
The data presented here are important for this field of Biological Degradation and Conversion, using extremophiles and deserves publication in Fermentation in my opinion, after some revision.
The paper is well written and in a comprehensive form. The Abstract is well constructed as the Introduction. The Introduction is an excellent example of an outstanding state of the art in this field.
Minor thing in Introduction: in line 87, the name of the genus “Sulfolobus” is not in italic. Please italicize it (Sulfolobus).
At the end of the Introduction, I would rephrase the last paragraph in order to make clear the objectives of your study. It is just a suggestion (lines 137-139).
Methods are well explained, but for the analytical techniques (section 2.3), I would explain these better, instead of indicating only the references. It is just a suggestion.
On the other hand, I liked very much the way the authors presented the Experimental system by a nice and clear schematic representation (section 2.2)
Just some more comments on the Methods (see 2.1). Where did you obtain the strain of your microorganism? From a culture collection? Was it isolated by you? This should be addressed. Is the strain identified with no doubts? By what methods of identification?
Results are well described and supported by a good number of tables and figures. The figures are of good quality. Statistical analyses were done comparing “couples” (pairs). Just a question: Why did you not perform an ANOVA among all treatments followed by a post-hoc test instead?
Line 252. It should be 3.2 and not 3.3 section (Study of the phosphorous limitation).
Discussion is well written and constructed. Nevertheless, the first 3 paragraphs need to be supported with some references in my opinion (lines 330-350). The same happens with the last 3 big paragraphs at the end of the Discussion (lines 361-384). In summary, the discussion needs to be much more supported by the literature, adding more references. The authors only use three references in the Discussion. By the way, the reference [46] is not referred in the text. I did not find it!
The References are well formatted, and many are updated. I also understand the use of some historical and old references. But the paper (especially the Discussion section) must be supported by more references (preferentially updated references).
Author Response

(The authors gave the same response as above.)

Round 2
Reviewer 1 Report
Title: Effect of the availability of the source of nitrogen and phosphorus in the bio-oxidation of H2S by Sulfolobus metallicus
The authors have replied most of the queries. Somehow, I consider that an extra work must be dedicated to improving the final text version. Concretely, I still find that discussion could be improved, and some sentences still need to be clear for the reader.
Some examples:
Abstract, lines 22-25.
“At the same time, sequential oxidation of various sulfur compounds according to their oxidation state was observed, predominant the products and the final oxidation of the sulfate ion, resulting from the oxidation of hydrogen sulfide, including the elemental sulfur and tetrathionate formed.”
Please clarify the ideas of this sentence.
Figure 1: Please, locate in the figure lanes 1-4.
Introduction
There are several paragraphs insisted on the need of kinetics (lines 85-102). However, there is no kinetic data in the article (data cannot be adjusted to a curve, and kinetic parameters cannot be provided). Moreover, these ideas are also suggested in lines 159-165. Please, reorganize these paragraphs to be more consistent.
Discussion must be revised and better focused. For instance, some ideas seem redundant, and it is not properly organized. Some examples:
Lines 658-666:” which indicates that it is an intermediate compound from the oxidation of the sulfide ion as a source of energy. Tetrathionate is an intermediate compound formed during the oxidation of sulfide ions by Sulfolobus metallicus.”
The second sentence seems to be redundant. Moreover, there are some figures/graphical that explain the S metabolism in the Sulfolobus strains as the figure 1 of your reference 29 (Front. Microbiol., 14 October 2021
Sec. Microbiotechnology
Volume 12 - 2021 | https://doi.org/10.3389/fmicb.2021.768283). Figure 1 from https://doi.org/10.1111/1462-2920.14712 can also be useful. I suggest authors to use this figure or a simpler one to explain why you centered your study on these compounds. Moreover, reference 29 and 47 are the same.
It would be useful to include a brief discussion using the reference https://doi.org/10.1111/1462-2920.14712 (https://ami-journals.onlinelibrary.wiley.com/doi/10.1111/1462-2920.14712).
Lines 680-684 related to phosphorus have a relationship again with lines 739-747. This is a bit confusing for the reader as the ideas are similar. Please, organize these ideas.
Line 689: “Cells of coccoid morphology with certain irregularities were observed in culture (Figure 5), with an average size of 0.9 μm.”
More information is needed for this figure, where is the control? How can we appreciate the differences respect to normal conditions? On the other hand, these stressful conditions are not mentioned in the legend of Figure 5. No mention of this technique or conditions done is mentioned in MM.
On the other hand, why this result is in discussion and not in result section?
Lines 748-751: “The oxidation curves of Sulfolobus metallicus show sequential oxidation of various sulfur compounds based on their oxidation state, with the predominant product being the sulfate ion, which is the final product of the oxidation of hydrogen sulfide. This process includes the formation and oxidation of elemental sulfur and tetrathionate. [54]”
Please, when you mention to “oxidation curves”, what are you referring to? Most of the graphics are not showing a curve at all (in the sense that you can make kinetics with it). On the other hand, I cannot see the “sequential oxidation” clearly stated. Once more, a graphic of the relationship among these compounds will help. Anyhow, this sentence here after the Phosphorus paragraph is a bit strange.
Minor changes:
Please, revise verb tenses. In many places you use the present when the past form would be more appropriate.
Please revise italics or capital letters when needed. Some examples (but not all) are:
e.g. Legend of Fig 5.
e.g. Line 827 On the Response of Halophilic Archaea to Space Conditions (the rest of references are not capitalized In the same way; some references are but not all, please check references style for this journey). Other references: ref 8, 12, 15 etc.
e.g. Line 891-2: acidianus ambivalens, acidianus infernus, stygiolobus azoricus, sulfuracidifex metallicus, and sulfurisphaera ohwakuensis.
E.g. Line 905 Escherichia coli

Author Response
Attached you will find the answers to the reviewer 1 observations.

Reviewer 2 Report
Comments:
I do thank all the efforts done by the authors to turn their article more comprehensive, and in this way to turn it into a better version of the initial one.
All my major concerns and questions were properly addressed and responded.
The methods are now much more well explained and the discussion was improved. Nevertheless, the Discussion still needs more literature support in my opinion. Now, there is a couple more of references in the discussion, but the number has raised only from 3 references (first version) to 7 (this version). I may understand that this is a upgrade, but at least for the two first paragraphs, and also for the paragraph before the last, of the Discussion, I would add more a couple of references to support what is said.
Nevertheless, I repeat that the paper was much improved in my opinion.
Just a brief comment or two.
I am satisfied with the methodology used for the identification of the microorganism that the authors used (and also making proof that it was not contaminated with anything else) but this could have been easier by DNA sequencing.
On the question about the using of t-tests and not ANOVA, the authors could have also used non-parametric ANOVA even after data transformation would not achieve the ANOVA assumptions. Nevertheless, I can accept the statistical analyses used by the authors.
Many thanks.
Author Response
Attached you will find the answers to the reviewer 2 observations.

Round 3
Reviewer 1 Report
Minor changes:
English style must still be polished (there are some words in plural that they should be singular and so on)
References must be carefully revised as there are different formats among them:
Examples: reference 5,8,13,17,19,27,29,42,43,46,47 etc. all the words of the title of the reference are capitalized and they must not be.
Figure 5 needs more explanation. In which growth condition has been this picture taken? Legend should be expanded to explain the conditions in which the picture was taken. Please point out in the picture the cells that must be taken as “normal” phenotype and those that can be considered as different from that pattern.
Author Response
Dear Editor:
Thanking the reviewers for their thoroughness and dedication to our work, I will now detail the respective responses to them:
- English style must still be polished (there are some words in plural that they should be singular and so on).
- A grammatical revision of the text was made, and some details of some singular words were found, and other grammatical details were corrected.
References must be carefully revised as there are different formats among them:
Examples: reference 5,8,13,17,19,27,29,42,43,46,47 etc. all the words of the title of the reference are capitalized and they must not be.
- All references were reviewed and corrected according to the format required by the journal.
Figure 5 needs more explanation. In which growth condition has been this picture taken? Legend should be expanded to explain the conditions in which the picture was taken. Please point out in the picture the cells that must be taken as "normal" phenotype and those that can be considered as different from that pattern.
- The detail was indicated with arrows in the figure, and a better and brief explanation of it was included.
We would like to thank you again for your comments on improving the work, and we hope they will be well received.